# Surface Modification by Nano-Structures Reduces Viable Bacterial Biofilm in Aerobic and Anaerobic Environments

**DOI:** 10.3390/ijms21197370

**Published:** 2020-10-06

**Authors:** Sarah Ya’ari, Michal Halperin-Sternfeld, Boris Rosin, Lihi Adler-Abramovich

**Affiliations:** 1Department of Oral Biology, The Goldschleger School of Dental Medicine, Sackler Faculty of Medicine, Tel Aviv University, Tel Aviv 6997801, Israel; sarahkal@gmail.com (S.Y.); michal4@mail.tau.ac.il (M.H.-S.); borisrosin@mail.tau.ac.il (B.R.); 2The Center for Nanoscience and Nanotechnology, Tel Aviv University, Tel Aviv 6997801, Israel; 3The Center for the Physics and Chemistry of Living Systems, Tel Aviv University, Tel Aviv 6997801, Israel

**Keywords:** modified amino acid, self-assembly, surface coating, anti-biofilm activity

## Abstract

Bacterial biofilm formation on wet surfaces represents a significant problem in medicine and environmental sciences. One of the strategies to prevent or eliminate surface adhesion of organisms is surface modification and coating. However, the current coating technologies possess several drawbacks, including limited durability, low biocompatibility and high cost. Here, we present a simple antibacterial modification of titanium, mica and glass surfaces using self-assembling nano-structures. We have designed two different nano-structure coatings composed of fluorinated phenylalanine via the drop-cast coating technique. We investigated and characterized the modified surfaces by scanning electron microscopy, X-ray diffraction and wettability analyses. Exploiting the antimicrobial property of the nano-structures, we successfully hindered the viability of *Streptococcus mutans* and *Enterococcus faecalis* on the coated surfaces in both aerobic and anaerobic conditions. Notably, we found lower bacteria adherence to the coated surfaces and a reduction of 86–99% in the total metabolic activity of the bacteria. Our results emphasize the interplay between self-assembly and antimicrobial activity of small self-assembling molecules, thus highlighting a new approach of biofilm control for implementation in biomedicine and other fields.

## 1. Introduction

Microbial adhesion and the subsequent formation of a biofilm on surfaces in a liquid environment is a natural phenomenon. However, these infections are difficult to detect [1,2], comprising a significant concern in different fields ranging from medical devices, surgical equipment, biosensors, water distribution systems [3,4], food storage [5,6] and industrial and marine instruments [7,8]. The commonly applied strategies to combat biofilm formation involve the prevention of initial bacterial adhesion to surfaces and biofilm degradation. The former mainly employs surface modifications, bactericidal coatings or anti-adhesive compounds as physical barriers [9,10], and the latter involves antimicrobial agents that kill or inhibit the growth of microorganisms [11]. Despite the advancements in the development of antimicrobial and anti-biofilm materials, coating techniques bear significant deficiencies, including the lack of long-term activity and stability, lack of coating adaptability to diverse materials, easy production and simple application [12].

Molecular self-assembly has attracted extensive interest in the construction of antimicrobial and antifouling agents [13,14,15,16,17,18,19,20,21,22,23,24]. Self-assembly is a spontaneous process in which the molecular organization of diverse building blocks, including nucleic acids, peptides, proteins and lipids, organize into well-ordered structures at the nano-scale [25,26,27,28]. It arises through non-covalent interactions between components, including electrostatic, hydrophobic, hydrogen bonding and π-π interactions [29]. Depending on the specific building blocks and the assembly conditions, various nano-structured morphologies, including fibrils, tubes, sheets, tapes, spheres, vesicles and hydrogel matrices, can be formed in vitro, allowing for distinctive functional possibilities [28,30,31,32,33,34,35], mainly in tissue engineering [36,37,38,39], regenerative medicine [40], cell culture [41,42,43,44], drug delivery [36,39,45,46,47,48,49], bio-imaging [50,51] and fabric functionalization [52]. Coatings based on self-assembled peptides, primarily for antibacterial purposes, have been recently reported [5,12,13,21,53,54,55,56]. The self-assembly of the tri-peptide DOPA-Phe(4F)-Phe(4F)-OMe into an antifouling coating by the dip-coating technique was reported [13]. The peptide-based coating has been shown to successfully inhibit *Pseudomonas aeruginosa* and *Escherichia coli* biofilm formation on titanium surfaces and was found to be biocompatible [13,56]. In another study, the combination of self-assembled polydopamine nano-aggregates with an ultrahigh molecular weight hydrophilic polymer had been utilized as a broad-spectrum antibacterial coating of titanium and polymeric surfaces [12]. Other studies utilized peptide self-assembly into coatings to improve cell attachment and proliferation [54,55]. For example, Dopa- and REDV-conjugated peptide amphiphiles have been used to functionalize stainless steel stent surfaces in order to enhance the attachment of endothelial cells aiming to enhance the long-term success of stents implantation [54].

Although peptide nano-structures have been the focus of many studies aimed towards technological applications, amino acids are gaining considerable interest as the simplest biological building blocks due to their commercial availability, low-cost, straightforward preparation and modification and biocompatibility [57]. Indeed, in recent years, much progress has been made in the generation of nano-structures from short and ultrashort peptides [58,59,60,61,62,63,64,65,66]. In 2012, phenylalanine (Phe), a single amino acid, was shown to form ordered fibrillary assemblies with amyloid-like properties due to both its aromatic structure and its hydrophobicity [67]. Later studies reported the self-assembly of Phe as well as other unmodified [68,69,70] and modified [71,72,73] amino acids to form fibrillary nano-structures.

The modification of amino acids can provide the assemblies with various properties, including antibacterial traits. Recently, Song et al. developed biometallohydrogels based on Ag^+^-coordinated Fmoc-amino acids self-assembly (Fmoc-l-serine, Fmoc-l-aspartic acid, Fmoc-l-leucine, Fmoc-l-proline, Fmoc-glycine) [24]. These metallohydrogels have been shown to have a significant antibacterial effect against both Gram-negative (*Escherichia coli*) and Gram-positive (*Staphylococcus aureus*) bacteria in cells and mice by their interaction with the cell walls and membrane, resulting in the detachment of the plasma membrane and leakage of the cytoplasm [24]. In another study, Nilsson and coworkers designed the Fluorenylmethoxycarbonyl (Fmoc)-pentafluoro-phenylalanine (Fmoc-F_5_-Phe) [73], which was later studied for its antibacterial properties [74]. Fmoc-F_5_-Phe nano-assemblies were incorporated into a dental resin-based composite restorative material and demonstrated their biocompatibility along with an inhibitory effect on *Streptococcus mutans* (*S. mutans*) in solution, targeting the bacterial cell membrane. This modified amino acid is comprised of the Fmoc-Phe moiety that induces nano-structure formation and the fluoride moieties, which are utilized for their antibacterial activity [74]. The crystallographic structure of Fmoc-F_5_-Phe was recently deciphered, leading to an understanding of the molecular interaction of thermodynamically stable structures, including π-π interaction and hydrogen bonding [75]. Furthermore, Fmoc-Phe has recently shown to confer an antimicrobial effect against Gram-positive bacteria such as *Staphylococcus aureus* both as a hydrogel and in solution. The underlying mechanism of this activity involves invasion into the bacterial cell followed by a reduction of glutathione levels [76]. The antibacterial properties of nano-assemblies formed by diphenylalanine have also been studied recently [19,20]. These assemblies cause membrane disruption in an *E. coli* model [20]. In addition, Fmoc-F_5_-Phe has been studied not only for its antibacterial potential but also due to its ability to co-assemble with the Fmoc-Phe-Phe dipeptide to form a fibrous hydrogel with extraordinary mechanical properties [77].

Here, inspired by these nano-structures’ properties, we present an antibacterial modification of titanium, mica, glass and siliconized glass surfaces via the drop-cast coating technique using two self-assembling modified amino acid structures. We chose the minimal self-assembling building block, Phe, decorated with fluoride moieties known for their antibacterial properties and their hydrophobic nature. We further modified the amino acid with the Fmoc or tert-butyloxycarbonyl (Boc) protective groups. The topography, wettability and stability of the modified surfaces were characterized and the antimicrobial effect towards two facultative anaerobic bacterial strains, *S. mutans* and *Enterococcus faecalis (E. faecalis)*, grown under anaerobic and aerobic conditions, respectively, was studied.

## 2. Results and Discussion

The surfaces’ coating was designed based on phenylalanine modified on its N-terminus and decorated with fluoride moieties, known for their antibacterial properties and their hydrophobic nature. The N-terminus of the amino acid was modified with either Fmoc or tert-butyloxycarbonyl (Boc) protective groups. The Fmoc protective group endows strong driving forces to the self-assembly process, such as hydrogen bonding from the carbonyl group, aromatic and hydrophobic interactions from the fluorenyl ring and steric optimization from the linker, the methoxycarbonyl group [30]. When located at the N-terminus of diphenylalanine, the Boc modification can form both nanospheres and nanotubes under different conditions [78]. Fabrication of the Fmoc-F_5_-Phe coatings was performed using the drop-cast technique. The Fmoc-F_5_-Phe building blocks (Figure 1A) were first dissolved in hexafluoronisopropanol (HFIP), a solvent with high evaporation propensity, which does not affect the tested surfaces’ hydrophobicity or topography, verified by contact angle and SEM, respectively. Dissolving the modified amino acids in HFIP reverts them to their monomeric form. For the Fmoc-F_5_-Phe coatings, 10 mm diameter titanium, mica, glass and siliconized glass discs were coated by drop-casting the solution at a concentration of 30 µL/cm^2^. Solvent evaporation at room temperature resulted in the self-assembly of Fmoc-F_5_-Phe into ordered nano-structures, forming a continuous, homogeneous coverage of the surface. As confirmed by scanning electron microscopy (SEM) analysis, the layer, formed by a single drop-cast coating, was unstable in aqueous solution and did not remain adhered to the surfaces. Therefore, we have modified the drop-cast technique and added a complementary stage of surface heating to 60 °C while dropping the modified amino acid solutions throughout the evaporation process. This process was repeated thrice using Fmoc-F_5_-Phe to create homogeneous, stable coatings of the surfaces (Figure 1B). This heat-driven procedure prompted the evaporation of the solvent, thereby facilitating the self-assembly process.

SEM analysis was used to characterize the topography of the different disc surfaces following the modified drop-casting process and after their immersion in water heated to 37 °C to assess the coating stability. A continuous and uniform coating of Fmoc-F_5_-Phe nano-assemblies could be seen on all four surfaces, before and after the stability test (Figure 1C–F and Appendix A). Amorphous Fmoc-F_5_-Phe nano-structures were observed on the titanium disc (Figure 1C) and similarly on the siliconized glass discs (Figure 1D). The structures remained intact and displayed a more defined architecture of spikey fibers stemming from the center after the stability test for titanium surfaces (Figure 1E). The same trend was observed on the siliconized glass surface (Figure 1F). These fibrils formed 10–20 µm buds, stemming from central granulation points. These fibrils appeared as sharp spikes and seemed accentuated and elongated in the connecting area between the buds. Fmoc-F_5_-Phe deposited on glass formed short intertwined fibers 10–20 µm in length and 1 µm in width (Appendix A) and on mica fibers 10 µm in length and 0.5 µm in width, stemming from nucleation points throughout the surface, could be observed (Appendix A). The structures on the glass surface maintained their morphology and dimension after the stability test (Appendix A). Fmoc-F_5_-Phe after the stability test on mica showed elongated fibers with spikes at the edges (Appendix A). Fmoc-F_5_-Phe coating stability was validated by X-ray diffraction (XRD) analysis. Crystalline structures with a similar pattern of diffraction peaks were observed both following the coating process and after the stability test (Figure 1G).

Moreover, we studied the Boc-F_5_-Phe (Figure 2A,B) coating on titanium, mica, glass and siliconized glass surfaces. Boc-F_5_-Phe forms a network of fibrils, approximately 0.5 µm in diameter, with a high aspect ratio on all four surfaces (Figure 2C,D and Appendix A). Upon immersion in 37 °C water for the stability test, the fibrils underwent a morphological transformation into larger rod assemblies, approximately 2 µm in diameter (Figure 2E,F and Appendix A). XRD analysis showed that Boc-F_5_-Phe initially formed an amorphous structure showing a broad indistinct peak and after wetting, the rods demonstrated a crystalline structure with distinct sharp peaks in the XRD spectra (Figure 2G).

In addition, contact angle analysis of a water droplet was used to analyze the surface wettability of the modified surfaces (Figure 3A–D). Hydrophobicity was significantly increased using both Fmoc-F_5_-Phe and Boc-F_5_-Phe on titanium, mica and glass surfaces compared to the unmodified surfaces (Figure 3E). The contact angle of each surface in different areas was consistent, indicating the uniformity of the coating. The highest change in hydrophobicity was observed on the mica surface. The measured water droplet contact angles on the Fmoc-F_5_-Phe and Boc-F_5_-Phe modified mica surfaces were θ = 80° and θ = 70°, respectively. These values represent a significant increase compared to unmodified mica, where the water droplet instantly covered the entire surface and the angle could not be measured. On the titanium-modified surfaces, a 3.5-fold and over 2-fold increase in hydrophobicity was obtained using Fmoc-F_5_-Phe (θ = 105°) and Boc-F_5_-Phe (θ = 70°), respectively, compared to the unmodified surface (θ = 30°) (Figure 3C,D). A similar trend of increased hydrophobicity was observed on the Fmoc-F_5_-Phe and Boc-F_5_-Phe glass surfaces, however, no significant change was observed on the modified siliconized glass surface, which exhibited a high contact angle of 80°, probably due to its inherent relatively hydrophobic nature (Figure 3E). 

Figure 3F,G shows side-view SEM images of the nano-structures deposited on siliconized glass. The Boc-F_5_-Phe coating was approximately 2.8-fold thicker than the Fmoc-F_5_-Phe coating. The measured thickness of the Boc-F_5_-Phe and Fmoc-F_5_-Phe layers was 55–60 µm (Figure 3F) and 17–22 µm (Figure 3G), respectively. The long-term stability of the coatings was tested in phosphate-buffered saline (PBS) and brain heart infusion (BHI) in order to assess their implementation in different areas of antibacterial treatment. The Fmoc-F_5_-Phe modification maintained a visible white coverage over the surface as opposed to Boc-F_5_-Phe, which proved to be unstable under the conditions required for bacterial growth. Therefore, only the Fmoc-F_5_-Phe modification was further tested for its antibacterial properties.

The antibacterial properties of the Fmoc-F_5_-Phe modified surfaces were analyzed using two bacterial strains. *E. faecalis* a facultative, Gram-positive bacterium known as a primary etiological agent of nosocomial infections and as a habitant of the oral cavity, specifically in filled root canals of teeth associated with apical periodontitis [79,80]. *S. mutans*, a Gram-positive, facultative anaerobic bacterium and one of the primary causes of human dental caries [81], which is also associated with bacteremia and infective endocarditis [82]. *E. faecalis* was grown in BHI broth containing 5% sucrose under aerobic conditions and formed a biofilm after 24 h incubation at 37 °C. The standard bacterial staining such as crystal violet and MTT (3-[4,5-dimethylthiazole-2-yl]-2,5-diphenyltetrazolium bromide) assays were inadequate for evaluating the biofilm on coated surfaces, since these dyes also stain the Fmoc-F_5_-Phe coatings, causing a very high background and no distinction between the bacteria and the coating (Appendix A). In order to overcome this obstacle, the surfaces were prepared for HRSEM imaging. The samples were dried and imaged by HRSEM. These images showed that areas of the Fmoc-F_5_-Phe-coated sample contain a reduced amount of adhered *E. faecalis* bacteria in comparison to the non-coated siliconized glass (Figure 4A,C). The areas covered with bacteria in Figure 4 are marked in green, the unmarked images are presented in Appendix A.

Furthermore, the biofilm of *S. mutans* was grown under anaerobic conditions in BHI broth containing 5% sucrose and bacitracin, an antibiotic *S. mutans* is resistant to, for 4 h at 37 °C. Modified and unmodified surfaces were placed in the *S. mutans* suspension for an additional 24 h. HRSEM analysis demonstrated the presence of adhered *S. mutans* bacteria on the coated siliconized glass surfaces (Figure 4B,D).

Next, a quantitative analysis of the biofilm viability of both *E. faecalis* and *S. mutans* on the modified surfaces compared to non-modified surfaces were analyzed (Figure 5). All four different surfaces coated with Fmoc-F_5_-Phe showed a significant reduction in *E. faecalis* viability, as indicated by attenuated ATP production, when compared and normalized to their corresponding non-modified surfaces (Figure 5A). A substantial reduction of approximately 94% was observed on mica and glass surfaces. The prominent reduction observed on the mica surface may be correlated to the significant change in surface wettability from extremely hydrophilic to hydrophobic. A significant reduction of approximately 86% and 87% in bacterial viability was also demonstrated on titanium and siliconized glass, respectively.

Notably, the biofilm of *S. mutans* also demonstrated a significant reduction of approximately 99% in ATP production on all four surfaces, highlighting the substantial antibacterial activity of the coating (Figure 5B). The two complementary methods that were used to study the bacteria interaction with the Fmoc-F_5_-Phe-coated surfaces enabled a comprehensive understanding of *E. faecalis* and *S. mutans* interaction with the surfaces. The qualitative electron microscopy methods were based on direct visualization of the bacteria on the surfaces together with the ATP quantification method for antibacterial properties assessment. The results demonstrate that the adherence of the bacteria is not prevented by the nano-structures’ coatings, however, the bacterial cell viability of the adhered bacteria is significantly reduced as demonstrated in the luminescence ATP assay.

In conclusion, we studied two phenylalanine-modified building blocks, Boc-F_5_-Phe and Fmoc-F_5_-Phe for various ordered-structures surface coatings. Although, both building blocks’ coatings increased the hydrophobicity of the titanium, mica, glass and siliconized glass surfaces, only the Fmoc-F_5_-Phe coating was stable and durable in aqueous solution. We further incubated the Fmoc-F_5_-Phe-coated surfaces with two bacteria strains under both aerobic and anaerobic conditions, demonstrating reduced *E. faecalis* adhesion and regular *S. mutans* adhesion. Notably, both bacteria stains showed a significant reduction in bacteria viability in luminescence ATP assay. In summary, we demonstrated the utilization of a simple self-assembling building block to form ordered structure for various surface coatings, demonstrating durability and antibacterial properties that can be useful in future biomedical applications.

## 3. Materials and Methods

### 3.1. Substrates

Four types of 10 mm diameter discs, namely mica (Electron Microscopy Sciences (EMS), Hatfield, PA, USA), titanium (William Gregor Ltd., Illford, UK), glass (Bar Naor Ltd., Ramat Gan, Israel) and siliconized glass (Hampton research, Aliso Viejo, CA, USA), were used. The modified amino acids, Fmoc-F_5_-Phe and Boc-F_5_-Phe (Sigma Aldrich, Rehovot, Israel), were separately dissolved by sonication in HFIP to a concentration of 50 mg/mL.

### 3.2. Coating Preparation Using the Drop-Cast Method

Prior to surface modification using the two modified amino acids, all discs were cleaned by sonication in 100% ethanol. The discs were then heated using a hot plate set to 60 °C. Amino acid solution (30 μL) was dropped onto the pre-heated substrate. The discs were heated for 30 min to allow solvent evaporation and self-assembly of the amino acids into nano-structures. Coatings were applied twice more using the same protocol.

### 3.3. Scanning Electron Microscopy (SEM)

Samples were dried under vacuum and then coated by a thin gold layer and viewed by SEM (JEOL, JSM-IT100 InTouchScope) or HRSEM (Zeiss, GeminiSEM 3000) with no additional coating.

### 3.4. Contact Angle Measurements

The static water contact angle measurements on the discs were performed using the Ramé-Hart goniometer with the Drop Image Advance analysis software. A 2 µL drop of DDW was deposited and the average contact angle was calculated from five measurements on each disc surface. The measurements were conducted at room temperature.

### 3.5. X-ray Diffraction (XRD)

Fmoc-F_5_-Phe and Boc-F_5_-Phe samples were prepared by the drop-cast method with heat-assisted adhesion on 20 × 20 mm glass slides and then analyzed by wide-angle XRD. The XRD pattern was collected using a Bruker’s D8 Discover Diffractometer; the used set-up was a θ:θ Bragg–Brentano geometry, the source was copper anode and the detector was a LYNXEYE XE linear detector. The diffraction patterns were collected between 4 and 40°2θ with step 0.02°2θ for 1 s per step.

### 3.6. Bacterial Strains and Growth Conditions

*S. mutans* (ATCC 35668) stored at −80 °C was transferred into BHI broth (Difco Brain Heart Infusion, 241830) supplemented with 5% sucrose (SIGMA-ALDRICH, sucrose for microbiology, ACS reagent, ≥99.0%, 84100) and Bacitracin (0.5 unit/mL) using an inoculating loop. The suspension was incubated at 37 °C in an air-tight container under anaerobic conditions using an anaerobe container system sachet (BD GasPak EZ anaerobe container system sachet, 260678). After four hours, the suspension was transferred to cover the samples in a 24-well plate and incubated for an additional 24 h under anaerobic conditions. 

*E. faecalis* (ATCC 29212) stored at −80 °C was transferred into BHI broth (Difco Brain Heart Infusion, 241830) supplemented with 5% sucrose (SIGMA-ALDRICH, sucrose for microbiology, ACS reagent, ≥99.0%, 84100) using an inoculating loop. The suspension was incubated at 37 °C overnight under aerobic conditions and then transferred to cover the samples in a 24-well plate and incubated for an additional 24 h at 37 °C under aerobic conditions.

### 3.7. Evaluation of Antibacterial Properties Using Luminescence Assay 

Quantification of microbial cell viability on the tested surfaces was based on bacterial ATP production which is used to convert beetle luciferin to oxyluciferin and light using the BacTiter Glo Microbial viability assay kit (Promega, G8231). The light emitted was quantified by Turner Biosystems Veritas Luminometer and Glomax 96 program and normalized to the corresponding non-coated surfaces treated with the same bacteria. All discs were treated with bacteria in growth medium, allowing for optimal conditions for bacterial growth and biofilm formation on the surfaces. After the allotted time, each surface was washed in PBS and transferred to a clean plate and immersed in 1.5 mL of fresh PBS. The quantitative assay was performed immediately.

## 4. Conclusions

In conclusion, we have demonstrated uniform, stable surface coatings of nano-structures formed by the self-assembly of simple building blocks. The characterization of these coatings by SEM, surface wettability and XRD provides an insight into their possible applications. The Fmoc-F_5_-Phe nano-structure coating caused a clear reduction in the metabolic activity of *E. faecalis* and *S. mutans*, as reflected by reduced ATP production, under both aerobic and anaerobic conditions, respectively. In addition, the coating slightly reduced the adherence of the *E. faecalis* and *S. mutans* to the surface. Therefore, the antibacterial properties of the Fmoc-F_5_-Phe structures that were previously demonstrated when incorporated into dental resin [74] are retained when drop-casted to form a thin antibacterial surface coating. This coating could potentially be used to fabricate dental appliances such as those used in the field of orthodontics and prosthodontics, to reduce the incidence of dental caries, thus improving oral health. The coatings showed a high durability in an aqueous environment, which is of utmost importance for such applications. 

## Figures and Tables

**Figure 1 ijms-21-07370-f001:**
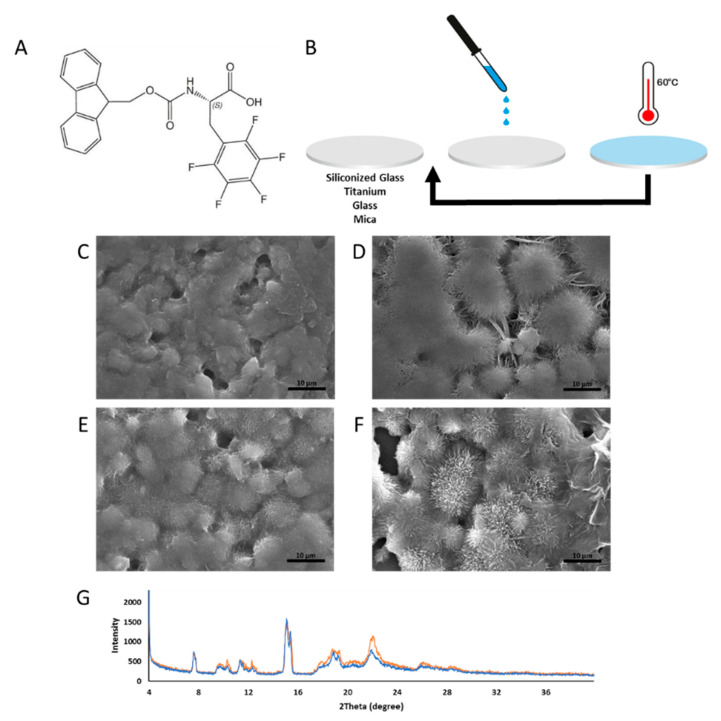
Characterization of the Fmoc-F_5_-Phe coating. (**A**) Molecular structure of the modified amino acid Fmoc-F_5_-Phe. (**B**) Schematic illustration of the modified drop-cast technique for surface coating of siliconized glass, titanium, glass and mica surfaces. The arrow represents the repetition of the drop-casting process and heating of the stages. (**C**,**D**) SEM structural characterization of Fmoc-F_5_-Phe-coated (**C**) titanium surface and (**D**) siliconized glass surface before stability test. (**E**,**F**) SEM structural characterization of Fmoc-F_5_-Phe-coated (**E**) titanium and (**F**) siliconized glass after stability test (Scale bar 10 µm). (**G**) XRD structural analysis of Fmoc-F_5_-Phe-coated glass surface before (orange) and after (blue) stability test.

**Figure 2 ijms-21-07370-f002:**
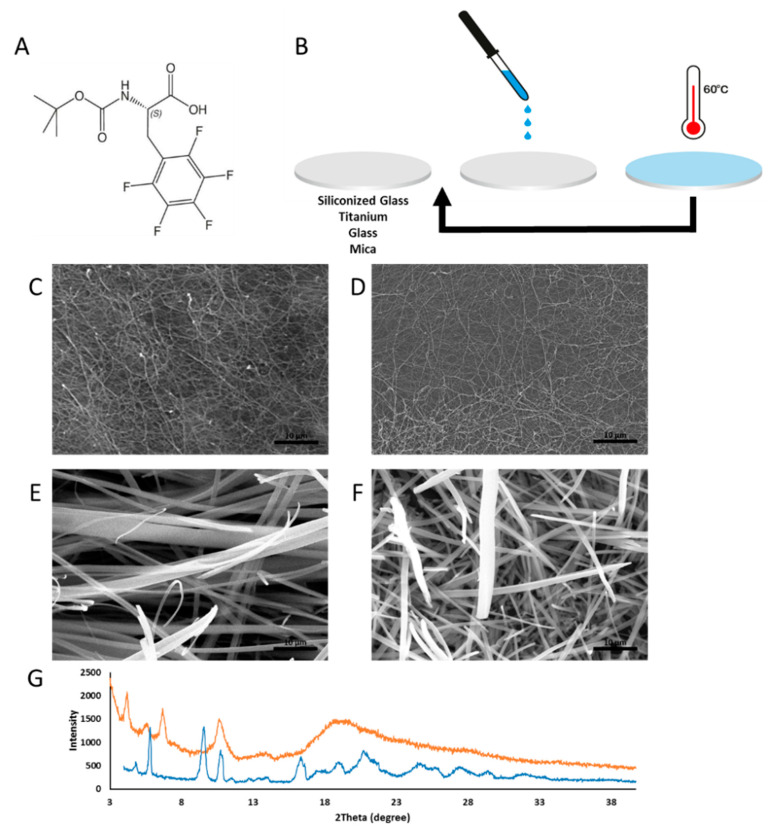
Characterization of the Boc-F_5_-Phe coating. (**A**) Molecular structure of the modified amino acid Boc-F_5_-Phe. (**B**) Schematic illustration of the modified drop-cast technique for surface modification of siliconized glass, titanium, glass and mica surfaces. The arrow represents the repetition of the drop-casting process and heating of the stages. (**C**,**D**) SEM structural characterization of Boc-F_5_-Phe-coated (**C**) titanium surface and (**D**) siliconized glass surface before stability test. (**E**,**F**) Boc-F_5_-Phe-coated (**E**) titanium and (**F**) siliconized glass after stability test (Scale bar 10 µm). (**G**) XRD structural analysis of Boc-F_5_-Phe-coated glass surface before (orange) and after (blue) stability test.

**Figure 3 ijms-21-07370-f003:**
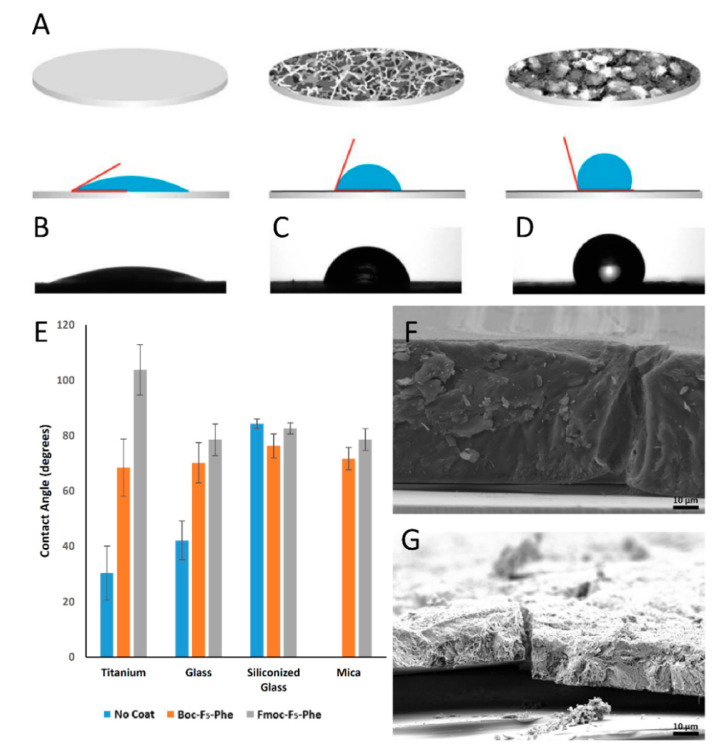
Surface properties of the Boc-F_5_-Phe and Fmoc-F_5_-Phe-coated discs. (**A**) Schematic view (top) and contact angle analysis of a 2 µL water droplet (bottom) of non-coated, Boc-F_5_-Phe-coated and Fmoc-F_5_-Phe-coated surfaces. The water droplet (blue), the black lines represents the coating on the surface and the red lines depict the contact angle. (**B**–**D**) Image of a water droplet on (**B**) a non-coated, (**C**) Boc-F_5_-Phe- and (**D**) Fmoc-F_5_-Phe- coated titanium surface. (**E**) Water contact angle measurements of non-coated, Boc-F_5_-Phe- and Fmoc-F_5_-Phe- coated titanium, glass, siliconized glass and mica (*n* = 3). (**F**) SEM image of the surface morphology and thickness of siliconized glass coated with Boc-F_5_-Phe (**G**) SEM image of the surface morphology and thickness of siliconized glass coated with Fmoc-F_5_-Phe (Scale bar 10 µm).

**Figure 4 ijms-21-07370-f004:**
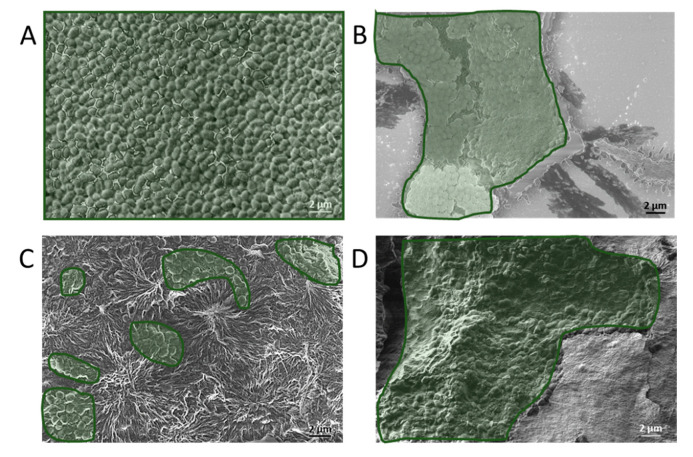
Biofilm analysis by HRSEM. (**A**) *E. faecalis* and (**B**) *S. mutans* form biofilm on the non-coated surface, (**C**) *E. faecalis* and (**D**) *S. mutans* incubated on Fmoc-F_5_-Phe-coated surface. The areas covered with bacteria are marked in green.

**Figure 5 ijms-21-07370-f005:**
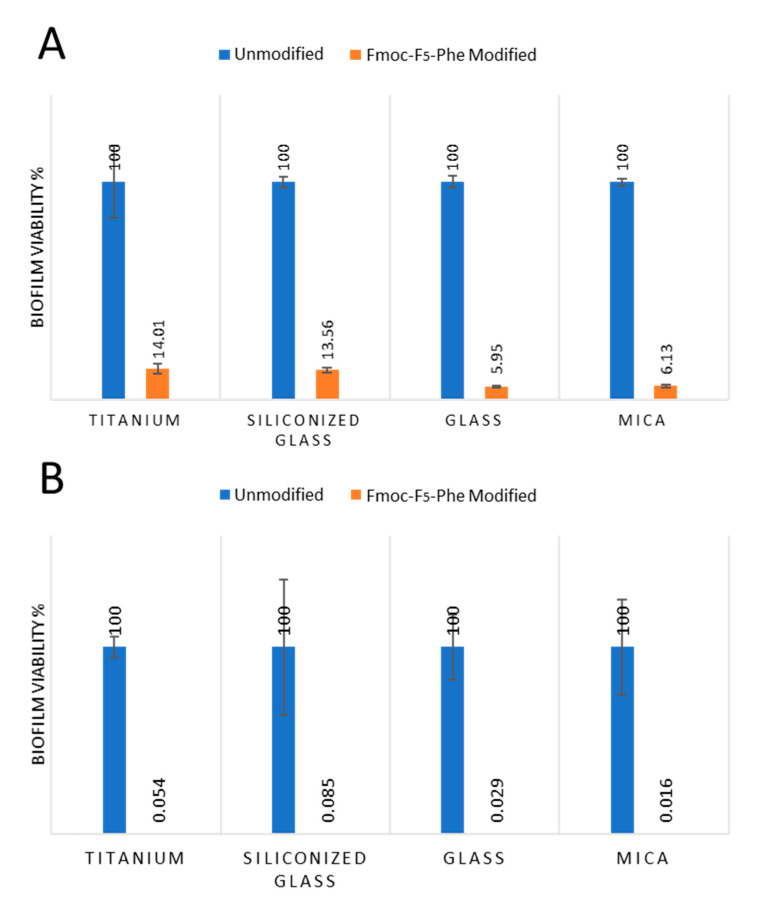
Quantification of biofilm viability on the modified surfaces. Relative (**A**) *E. faecalis* and (**B**) *S. mutans* biofilm viability on the four substrates modified by Fmoc-F_5_-Phe was compared to the corresponding non-modified surfaces (*n* = 3). *p*-Test < 0.05 for all substrates.

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
