# Peer review of "Surface Modification by Nano-Structures Reduces Viable Bacterial Biofilm in Aerobic and Anaerobic Environments"

_ijms, 2020, doi:10.3390/ijms21197370_

Round 1
Reviewer 1 Report
The authors present an investigation on the antibacterial properties of mica, glass and titanium surfaces coated with Fmoc-F5-Phe nanostructure assemblies. This work is motivated by and is an extension of their earlier work on the incorporation of the nano assemblies into dental resin based composites. The reported work falls within the scope of the journal and can be considered for publication after attention to the following:
- Line 111. It is evident that success was achieved at 60°C but what conditions were used for the initial solvent evaporation?
- What effect does the solvent have on the substrate surfaces?
Author Response
Reviewer 1
The authors present an investigation on the antibacterial properties of mica, glass and titanium surfaces coated with Fmoc-F5-Phe nanostructure assemblies. This work is motivated by and is an extension of their earlier work on the incorporation of the nano assemblies into dental resin based composites. The reported work falls within the scope of the journal and can be considered for publication after attention to the following:
We thank the reviewer for the positive and encouraging assessment of this study and for providing valuable suggestions that have assisted us in revising the manuscript to include additional details.
Special comments for the revision:
- It is evident that success was achieved at 60°C but what conditions were used for the initial solvent evaporation?
We thank the reviewer for the opportunity to clarify the experimental conditions. The solvent was originally allowed to evaporate at room temperature before the ideal temperature of 60° C was tested. We added additional information into the manuscript (Page 3, line 111):
"Solvent evaporation at room temperature resulted in the self-assembly of Fmoc-F5-Phe into ordered nanostructures, forming a continuous, homogeneous coverage of the surface. As confirmed by scanning electron microscopy (SEM) analysis, the layer, formed by a single drop-cast coating, was unstable in aqueous solution and did not remain adhered to the surfaces. Therefore, we have modified the drop-cast technique and added a complementary stage of surface heating to 60Ëš C while dropping the modified amino acid solutions and throughout the evaporation process. This process was repeated thrice using Fmoc-F5-Phe to create homogeneous, stable coatings of the surfaces (Figure 1B)."
- What effect does the solvent have on the substrate surfaces?
We thank the reviewer for highlighting this issue. These surfaces are unaffected by the solvent. We verified this by performing both contact angle and electron microscopy for non-coated surfaced exposed only to the solvent under similar conditions to those forming the nanostructures. We have now clarified this in the text of the manuscript, lines 109-110:
"The Fmoc-F5-Phe building blocks (Figure 1A) were first dissolved in hexafluoronisopropanol (HFIP), a solvent with high evaporation propensity, which does not affect the tested surfaces hydrophobicity or topography, verified by contact angle and SEM, respectively. "
Reviewer 2 Report
The present paper by Ya‘ari et al describes the attempt on a dedicated surface modification by the self-assembling of fluorinated phenylalanines on various substrates to create a protecting environment against bacterial biofilm formation. The paper describes a versatile method for spreading and attaching the modified amino acids to the various surfaces and demonstrate a reduced metabolic activity of attached bacterial biofilms to these modified surfaces. The paper is well written with an extensive overview on the subject in the Introduction and a good description of the methods to proof the concept and the antimicrobial activity. The paper might be published by Molecular Structure with some minor modifications, also the reviewer tends to suggest to publish the paper in a journal more devoted to biomedical surface modifications and biological coatings.
The authors may comment on the following points to further improve the manuscript.
- 1, line 43: It should read ‘… hydrogen bonding …’, not ‘hydrogen bonds’
- 2, line 73: The bacteria names should be in italics and the name ‘Gram-‘ should be written ‘gram-‘
- 4, caption Figure 1: the description of part G of the figure regarding the XRD pattern is missing.
- 1, line 136: The authors claim to have validated the coating stability by XRD. The authors have shown structural changes by XRD which is not automatically related to the stability of the nano-structure covering the surfaces. The authors also give no interpretation of the XRD structure in terms of found structural arrangement of the self-assembled amino acids by crystallographic means. As the found diffraction peaks indicate a structural arrangement the authors should give such a description of the structure presented.
- 7, line 183: the authors discuss the HRSEM images do show areas with reduced amount of adhered bacteria. This is difficult to validate by the quality of the micrographs without statistical analysis. The authors should highlight in the pictures the relevant areas and estimate the reduced covered space.
- 8, line 216: It should be read ‘… properties that can be useful in …’
- The authors have applied mainly electron microscopy to describe the surface coating. Did the authors also try cryo-TEM and AFM?
- 10, line 275: The authors use the term ‘antibacterial nano-coating’. Although the term ‘nano-coating’ may sound fancy, the procedure described is a self-assembly of organic molecules to a thin surface layer of nano- to micrometer thickness. The authors should be careful in describing the interesting procedure for a reliable surface coating by the procedure presented with wrong expectations of a nano-structured surface modification.
